# Surface Modification of Mg0.8Ca Alloy via Wollastonite Micro-Arc Coatings: Significant Improvement in Corrosion Resistance

**Mariya B. Sedelnikova** [1,*], **Anna V. Ugodchikova** [2], **Tatiana V. Tolkacheva** [1], **Valentina V. Chebodaeva** [1], **Ivan A. Cluklhov** [1], **Margarita A. Khimich** [3], **Olga V. Bakina** [3], **Marat I. Lerner** [4], **Vladimir S. Egorkin** [5], **Juergen Schmidt** [6] and **Yurii P. Sharkeev** [1,7]

[1] Laboratory of Physics of Nanostructured Biocomposites, Institute of Strength Physics and Materials Science of SB RAS, 634055 Tomsk, Russia; tolkacheva@ispms.tsc.ru (T.V.T.); vtina5@mail.ru (V.V.C.); gia@ispms.tsc.ru (I.A.C.); sharkeev@ispms.tsc.ru (Y.P.S.)

[2] Laboratory of Plasma Synthesis of Materials, Troitsk Institute for Innovation & Fusion Research, 108840 Moscow, Russia; ugodchikova@triniti.ru

[3] Laboratory of Nanobioengineering, Institute of Strength Physics and Materials Science of SB RAS, 634055 Tomsk, Russia; khimich@ispms.tsc.ru (M.A.K.); ovbakina@ispms.tsc.ru (O.V.B.)

[4] Laboratory of Physical Chemistry of Ultrafine Materials, Institute of Strength Physics and Materials Science of SB RAS, 634055 Tomsk, Russia; lerner@ispms.tsc.ru

[5] Institute of Chemistry FEB RAS, 100-letiya Vladivostok Prospect 159, 690022 Vladivostok, Russia; egorkin@ich.dvo.ru

[6] Department of Electrochemistry, Innovent Technology Development, Pruessingstrasse 27 B, D-07745 Jena, Germany; JS@innovent-jena.de

[7] Research School of High-Energy Physics, National Research Tomsk Polytechnic University, Lenin Prospect 30, 634050 Tomsk, Russia

* Correspondence: smasha5@yandex.ru; Tel.: +7-3822-286-887

**Abstract:** Biodegradable materials are currently attracting the attention of scientists as materials for implants in reconstructive medicine. At the same time, ceramics based on calcium silicates are promising materials for bone recovery, because $Ca^{2+}$ and $Si^{2+}$ ions are necessary for the mineralization process, and they take an active part in the formation of apatite. In the presented research, the protective silicate biocoatings on a Mg0.8Ca alloy were formed by means of the micro-arc oxidation method, and the study of their morphology, structure, phase composition, corrosion, and biological properties was carried out. Elongated crystals and pores were uniformly distributed over the surface of the coatings. The coated samples exhibited remarkable anti-corrosion properties in comparison with bare magnesium alloy because their corrosion current decreased 10 times, and their corrosion resistance increased almost 100 times. The coatings did not significantly affect the viability of the cells, even without the additional dilution of the extract, and were non-toxic according to ISO 10993-5: 2009. In this case, there was a significant difference in toxicity of the pure Mg0.8Ca alloy and the coated samples. Thus, the results demonstrated that the applied coatings significantly reduced the toxicity of the alloy.

**Keywords:** micro-arc oxidation; magnesium alloy; biocoating; wollastonite; bioresorption; corrosion resistance; cytocompatibility

## 1. Introduction

In the modern world, the number of diseases associated with damage to human bone tissue caused by injuries, fractures, and congenital genetic abnormalities has sharply increased [1]. Population aging and an increase in the amount of overweight people exacerbate this trend. Therefore, meeting the need for treatment and human bone replacement has become an urgent clinical, scientific, and socio-economic problem [1,2].

The complicated bone healing process requires the use of biomaterials and the development of implant designs. Biologically active elements or factors are also needed in order to improve fracture healing [2]. Today, bioresorbable/biodegradable materials are of great interest as implant materials for reconstructive medicine, due to their ability to dissolve and/or be slowly replaced by progressive tissues, such as newly formed bones [2,3]. The property of bioresorbability in the body environment provides an opportunity to avoid repeated surgeries and, consequently, possible inflammation processes and complications. Bioresorbable materials include such types of materials as some polymers [4,5], bioglass [6–8], ceramics [8–10], and some metal alloys [11–14]. It has been noted that polymers demonstrate unsuitable mechanical properties; furthermore, the biocompatibility of polymers regarding the surrounding tissues turned out to be very ambiguous [5]. Many authors describe the excellent biocompatibility and osteoinductive properties of bioglass [6]. However, there is a contradiction between its mechanical properties and bioactivity [7,8]. Bioresorbable ceramic materials gradually decompose in the human body and are replaced by newly formed bone tissue. However, when ceramics dissolve, changes in body composition in the field of implantation can occur. Moreover, problems with the preservation of strength and stability throughout the period of degradation can take place [3,10].

Alloys of magnesium, zinc, and iron can be distinguished among bioresorbable metals. Zinc and iron have been actively mentioned in the literature as possible candidates for bioresorbable implants only in the last five years [11–13]. Using various alloying elements, zinc- and iron-based alloys with adjustable mechanical properties and corrosion rates have been obtained. Resorbable alloys can be used to create biomedical implants, such as cardiovascular and orthopedic devices [11,13]. It was revealed that, in terms of their mechanical properties (for example, fatigue strength), porous iron and zinc demonstrate higher values than porous magnesium alloy (WE43), which is explained by their higher ductility [12]. In addition, there is growing interest in Ti-, Mg-, and Fe-based multifunctional metallic biomaterials as bone substitutes [14].

Mg alloys as orthopedic biomaterials have huge advantages—such as a low weight and density, as well as an elastic modulus similar to that of human bone—which exclude the probability of shielding bone stress [9,15–17]. Magnesium is one of the main elements of human organisms. This element is actively involved in processes that are closely related to cell differentiation and the mineralization of calcified tissue, and indirectly affect mineral metabolism [16,17]. The disadvantage of magnesium is its weak strength properties. However, such mechanical characteristics as strength, fracture toughness, creep resistance, hardness, and biocompatible properties can be improved by alloying magnesium with different elements (Al, Zn, Ca, Mn, Sn, Ce, etc.) [15,16,18]. Mg–Ca alloys as bioresorbable/biodegradable materials provide excellent biocompatibility and sufficient strength. In medicine, it is especially appropriate to use calcium, since the decomposition products of such magnesium alloys are absolutely non-toxic [15].

Another drawback of magnesium alloys is their low corrosion resistance [19–21], especially under the strain of physiological loads. The internal environment of the human body (tissue fluid) is quite reactive because it contains sodium chloride, plasma, proteins, and amino acids [21]. The interaction between the corrosion products and the surrounding tissues is an important research aspect [16,17]. Corrosion products of magnesium alloys, including released alloying elements, and gaseous products such as $H_2$ and $OH^-$, have a significant effect on the material's biocompatibility. Usually, $Mg^{2+}$ ions are non-toxic and even, as mentioned above, useful for many biological processes in the human body [22].

The surface modification of biometals by coating is an effective method of improving their corrosive and biocompatible properties. The biocoatings, on the one hand, play the role of a protective anti-corrosion layer and, on the other, enforce the bioactive properties of magnesium-based implants [15,20–23].

Calcium phosphates such as hydroxyapatite, tricalcium phosphate and monetite have currently been the most commonly used materials to create biocoatings [24–27]. However, today, ceramic materials based on calcium silicates are considered more suitable

for bone tissue repair [28–42]. $Ca^{2+}$ and $Si^{4+}$ ions play a key role in apatite layer nucleation; moreover, they have considerable effects on the biological metabolism of the osteoblast cells, which are necessary for the bone fusion mechanism and for collagen mineralization [29,37]. Zhai et al. [36] discovered that Ca–Mg–Si-containing biomaterials could not only control osteogenic differentiation of mesenchymal stem cells, but also induce angiogenesis of endothelial cells.

Apatite layer nucleation occurs because of Ca ions' release from the surface and, as a consequence, the formation of many silane (Si–OH) groups [28,29]. Si-OH groups make the structure of apatite heterogeneous, and Ca ions enhance the ionic activity and nucleation of apatite. When apatite nuclei form on the surface, they spontaneously grow using calcium phosphate ions from SBF [34,39]. Wollastonite is the most popular type of silicate ceramic for biomedical applications. This material demonstrates excellent biocompatibility (without the formation of a fibrous layer between the implant and the bone) and good mechanical properties; however, it also shows a high dissolution rate [30–32]. Compounds such as diopside ($CaMgSi_2O_6$) [33–38], akermanite ($Ca_2MgSi_2O_7$) [37–39], bredigite ($Ca_7MgSi_4O_{16}$) [35,36], and glass–ceramic materials [40–43] in the $CaO–MgO–SiO_2$ system have become the subject of intense research. They exhibit more significant new bone regeneration than the calcium phosphate ceramics popular today (e.g., hydroxyapatite, β-tricalcium phosphate) [44]. In addition, the described compounds can be used to create coatings on the surface of magnesium alloys [45–48].

Micro-arc oxidation (MAO), also called plasma electrolytic or anode spark deposition, is a prospective method of modification of valve metals' surfaces via the formation of ceramic-like coatings [49,50]. MAO coatings can be used for various applications—for example, to form a hard and wear-resistant layer on aluminum, titanium, or magnesium alloys [50]; to create a chemically resistant anticorrosive layer [51–53]; or even for the coloring of metal surfaces [54]. By varying the parameters of the deposition process and the composition of the compounds included in the electrolyte, it is possible to form the coatings with a developed surface morphology and with high osteogenic properties [49,50]. Therefore, the aim of the present research was the synthesis of micro-arc coatings on the surface of a Mg0.8Ca alloy with the participation of natural wollastonite particles, as well as the investigation of how the structure, morphology, and phase composition of the coatings influence their corrosion resistance, behavior in biological fluid, and toxicity with respect to the 3T3 fibroblast cell line.

## 2. Materials and Methods

### 2.1. Sample Preparation

Mg–0.8 wt.% Ca (Mg0.8Ca) is the magnesium alloy, which was used as the material for the substrate. This alloy was produced at Helmholtz Zentrum (Geesthacht, Germany). We describe the process of the alloy's production elsewhere [27]. Samples from the Mg0.8Ca alloy were in the shape of a parallelepiped, with sizes of $10 \times 10 \times 1$ $mm^3$. Experimental samples were subjected to grinding with 1200 grit sandpaper containing silicon carbide. After this, the grinding samples were ultrasonically washed in distilled water for 10 min. Such treatment led to the roughness of the prepared samples being Ra = 0.3–0.6 μm.

The coatings were synthesized using the MAO method in the anodic potentiostatic regime for 5 min, under the applied voltages of 350–500 V, with the help of Micro-Arc 3.0 software, as reported in previous papers [27,31]. The electrolyte containing wollastonite ($CaSiO_3$), NaOH, $Na_2SiO_3$, and NaF was used for the coating's deposition. Wollastonite MIVOLL®05-96 (ZAO GEOKOM, Kaluga region, Russia) was used in the work. Endless chains of silicon–oxygen tetrahedra make up the structure of wollastonite. The $Ca^{2+}$ ions are located between the chains as if they were "stitching" them [55]. The recurrence period in wollastonite chains is three tetrahedrons (Figure 1a). Due to this structure, wollastonite crystals have an elongated shape, which is drawn-out along the chain (Figure 1b). In this study, wollastonite with the following particle sizes was used (L—length): average $L_{av}$, maximum $L_{max}$, and minimum $L_{min}$ equal to 35 μm, 130 μm, and 4 μm, respectively. The

chemical composition of wollastonite was as follows: wt.%: $SiO_2$ 50.0–53.0, CaO 45.0–48.0, MgO 0.4–1.0, $Al_2O_3$ 0.1–0.3, and $Fe_2O_3$ 0.05–0.2. Wollastonite of this kind is characterized by high purity, which is important when creating biocoatings for implants.

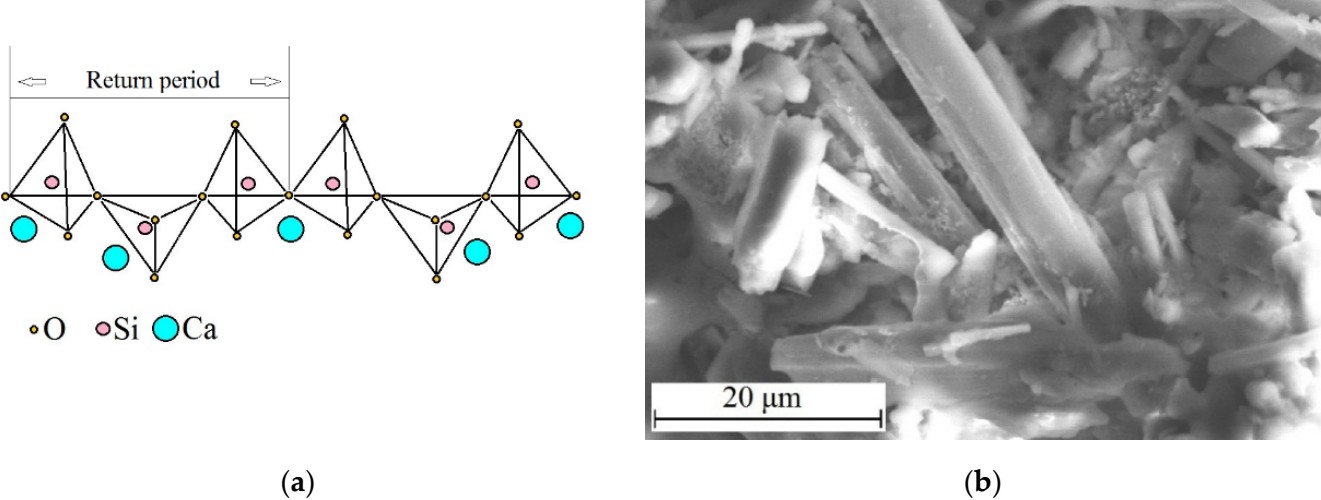

(**a**)　　　　　　　　　　　　　　　　　　(**b**)

**Figure 1.** Scheme of the wollastonite chain (**a**), and an SEM image of the wollastonite particles (**b**).

### 2.2. Experimental Methods

The coating's microstructure, morphology, and elemental composition were studied using an LEO EVO 50 scanning electron microscope (SEM, Zeiss, Jena, Germany) equipped with an energy dispersive X-ray spectroscope (EDX, INCA, Oxfrod Instruments, Abingdon, UK), and a JEM-2100 transmission electron microscope (TEM, Jeol Ltd., Musashino, Akishima, Tokyo, Japan) in the "Nanotech" Common Use Center of the Institute of Strength Physics and Materials Science Siberian Branch of the Russian Academy of Sciences (Tomsk, Russia). The surface roughness was estimated as average roughness (Ra) using a Hommel–Etamic T1000 profilometer (Jenoptic, Jena, Germany) at the National Research Tomsk Polytechnic University. The profile characteristics, such as traverse length and measuring rate, were 6 mm and 0.5 mm/s, respectively. The X-ray powder diffraction (XRD) method made it possible to determine the coating's phase composition (XRD, DRON-7, Burevestnik, Russia, "Nanotech" center at ISPMS SB RAS) in the angular range $2\theta = 5$–$90°$, with a scan step of $0.02°$ with Co K$\alpha$ radiation ($\lambda = 0.17902$ nm). The Database of the Joint Committee on Powder Diffraction Standards (JCPDS) was used for the phase identification and interpretation of X-ray profiles. The obtained XRD profiles were used to calculate the volume ratios of the crystalline and amorphous phases using Rietveld refinement. Fourier-transform infrared spectroscopy (FTIRS) was performed using an Alpha IR-spectrometer (Bruker, Karlsruhe, Germany) in reflection mode, in the wave number range of 1500–500 cm$^{-1}$.

### 2.3. Electrochemical Studies

The electrochemical properties of the pure alloy and coated samples were studied using a Versa STAT MC system (Princeton Applied Research, Oak Ridge, TN, USA). The measurements were carried out in a three-electrode cell K0235, with a 0.9% NaCl aqueous solution. This methodology is described in previous works [27].

The linear polarization resistance experiment was performed, progressing from $-30$ mV to 30 mV vs. OCP, at a scan rate of 0.167 mV/s. The polarization resistance, $R_p$, was calculated from the linear potential–current density plot as the $R_p = \Delta E/\Delta j$, as recommended in [56]. Potentiodynamic polarization curves were obtained at a scan rate of 1 mV/s, which is typical for Mg alloys, as opposed to 0.167 mV/s. The Levenberg–

Marquardt (LEV) method was used to fit the experimental polarization curve (potential, $E$, vs. current density, $j$) with the following Equation (1):

$$j = j_C(10^{(E-E_C)/\beta a} + 10^{-(E-E_C)/\beta_c}) \tag{1}$$

which gives the best fit values of corrosion potential, $E_C$, and corrosion current density, $j_C$ [57]. The measurements of the electrochemical impedance spectroscopy (EIS) were carried out using a sinusoidal perturbation signal with an amplitude r.m.s. of 10 mV. Versa STUDIO (Princeton Applied Research, Oak Ridge, TN, USA), ZView, and CorrView software (Scribner Associates, AMETEK, Mahwah, NJ, USA) were used for the experiment's control and analysis. Each experiment was performed on the three different samples.

*2.4. Biological Studies*

We studied the biodegradability of samples in biological fluids (0.85 wt.% NaCl), according to ISO 10993-5. The samples were immersed in the solution at 37 °C for 16 days. The samples' weight loss was calculated by the formula (2):

$$\Delta m = (m_0 - m_i/m_0) \times 100\% \tag{2}$$

where $m_0$ is the mass before the dissolution and $m_i$ is the mass after the dissolution. The mouse 3T3 fibroblast line was used for evaluation of toxicity (State Research Center of Virology and Biotechnology "VECTOR" Novosibirsk, Russia). Cells were grown in a DMEM medium supplemented with 2 mM L-glutamine (HyClone, Logan, UT, USA), 10% fetal bovine serum FCS (HyClone, USA), and 1% penicillin/streptomycin (HyClone, USA). The samples were extracted for 3 h at a surface/volume ratio equal to 1 cm$^2$/mL of cell medium. To assess the effect of particles on the viability of the cell lines, an MTT assay was performed. The incubation with MTT solution was carried out for 2 h at 37 °C and 5% $CO_2$. The optical density was determined on a Thermo Scientific Multiskan FC microplate spectrophotometer (Thermo Fisher Scientific, Waltham, MA, USA) at a wavelength of 570 nm. Parametric methods with a confidence level of $p \leq 0.05$ were used for statistical data processing. The determination of the pH of the cell medium was performed during the preparation of the extract within a 24 h period.

**3. Results**

*3.1. Morphology, Structure, and Elemental Composition of the W-Coatings*

An anodic potentiostatic mode was used for the W-coating deposition using the MAO method. The coatings were formed at fixed voltages equal to 350, 400, 450, and 500 V. The graphs of the current density at the time of coating deposition, recorded at different process voltages, are shown in Figure 2a. At the initial moment of the coating deposition, the current density reached its maximum and then decreased. The reason for this was an increase in the thickness of the ceramic-like coating. The decrease in the current density occurred at various rates for the different voltages. At a low voltage of 350 V, weak micro-arc discharges were realized in the system, and the current density dropped to a minimum value almost immediately, during the first 20 s of the process. At the voltages of 400–450 V, the current density decreased within 100 s according to the exponential law. At 500 V, the current density decreased evenly over the entire deposition period.

With the increase in the process voltage, the intensity of the micro-arc discharges increased and the temperature in the region of the micro-arc discharges rose. When the wollastonite was exposed to heat, at a temperature of 1125 °C, the polymorphic transition of β-wollastonite to α-wollastonite (pseudowollastonite) took place [30]. At a temperature of 1544 °C wollastonite melts [58]. In this case, ions passed into the system as charge carriers and then participated in the formation of micro-arc discharges. Mortazavi et al. [59] reported that the total applied current in this process consists of an electron current caused by sparking and an ion current caused by ion diffusion. Thus, a higher applied voltage led to a more intense melt formation in the system, and the MAO process continued for a

longer period. The increase in the voltage of the MAO process from 350 to 500 V led to the increase in the thickness and roughness of the coatings from 40 to 150 μm, and from 6.5 to 10.5 μm, respectively (Figure 2b). The surface morphology of the W-coatings' surfaces, and their cross-section image, are shown in Figure 3.

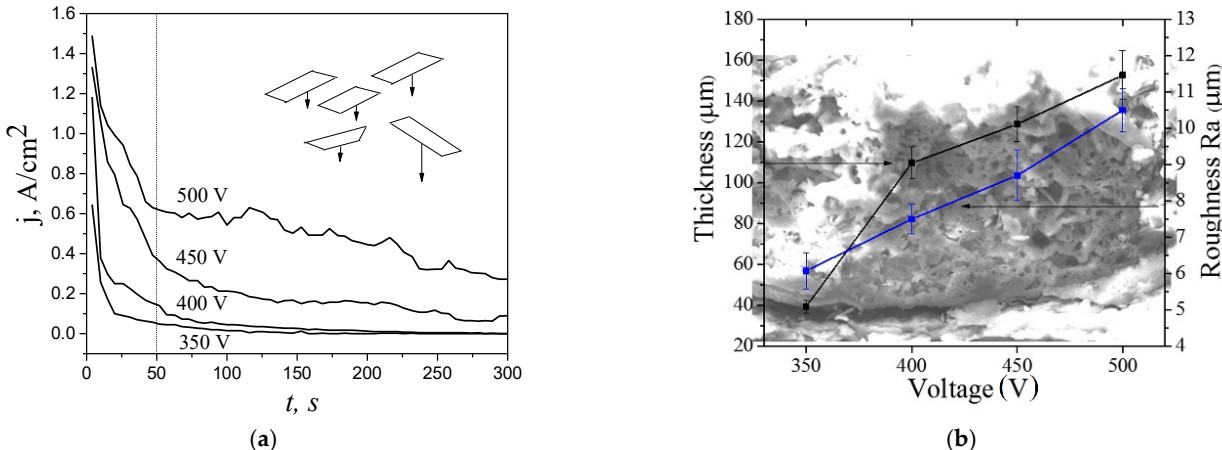

**Figure 2.** The plots of the current density versus the MAO process duration for the W-coatings' deposition (**a**), the plots of the coatings' thickness (black line) and Ra roughness (blue line) versus the applied voltage (**b**).

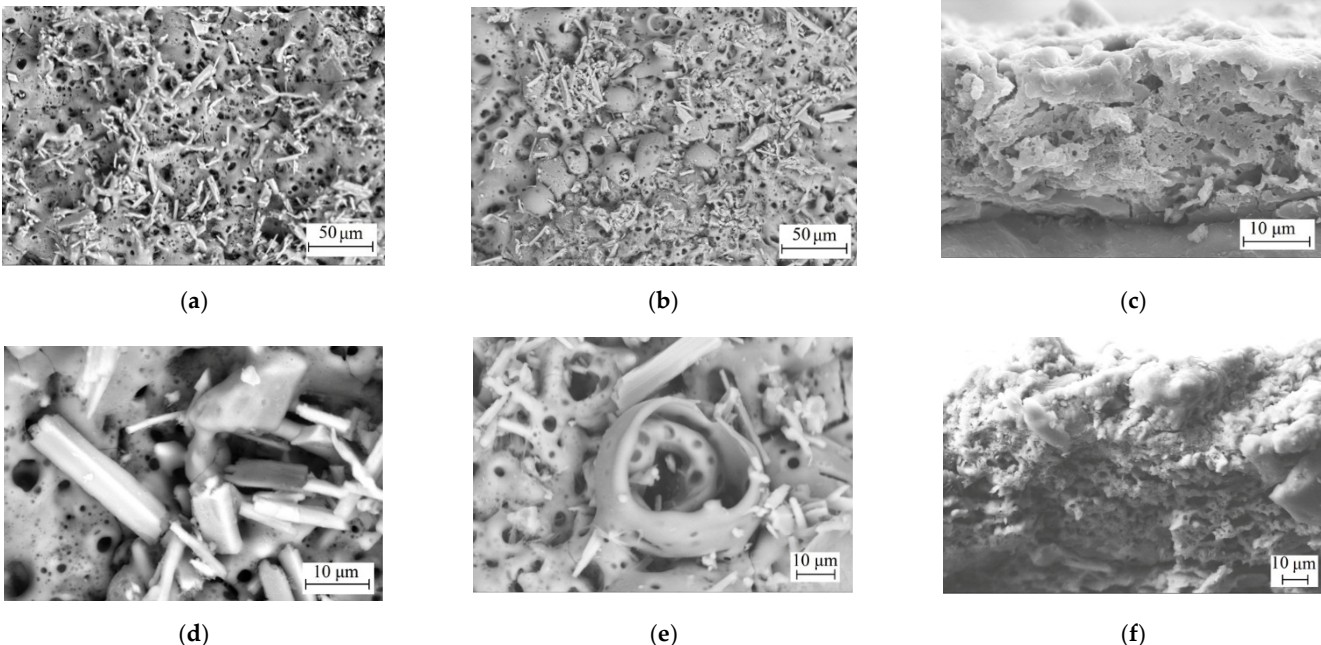

**Figure 3.** SEM images of the surface (**a**,**b**,**d**,**e**), and cross-sections (**c**,**f**), of W-coatings deposited at 350 V (**a**,**d**,**c**) and 500 V (**b**,**e**,**f**).

Elongated crystals can be seen on the coating's surface (Figure 3a,b,d,e). This form of crystal is common for natural mineral wollastonite. Moreover, pores with sizes of 2–7 μm are observed in the coatings. Spherical formations (bubbles) can be seen in the coating deposited at 500 V, in the field of the intense micro-arc discharge output. This "boiling" of the coating substance [60,61] is due to the melting of electrolyte components and the formation of the melt. SEM images of the cross-sections of the coatings show that the coatings have a loose porous structure (Figure 3c,f). The pores are evenly distributed throughout the coating's thickness. No wollastonite particles are observed inside the coating.

The results of the elemental analysis of the W-coatings are shown in Figure 4. Element distribution maps (Figure 4b) demonstrate that Si and Ca are concentrated mainly in the

areas of accumulation of the elongated wollastonite crystals, while Mg is observed in the areas free from the wollastonite particles. The energy spectrum shows (Figure 4c) that Si, Ca, and O elements on the coating's surface.

The element distribution tracks along the cross-section of the coating are shown in Figure 5. Mg and O prevail in the boundary between the coating and the magnesium substrate, which is explained by the oxide layer MgO formation. Closer to the surface layer, the amount of Mg increases again, as well as the amounts of Si and O.

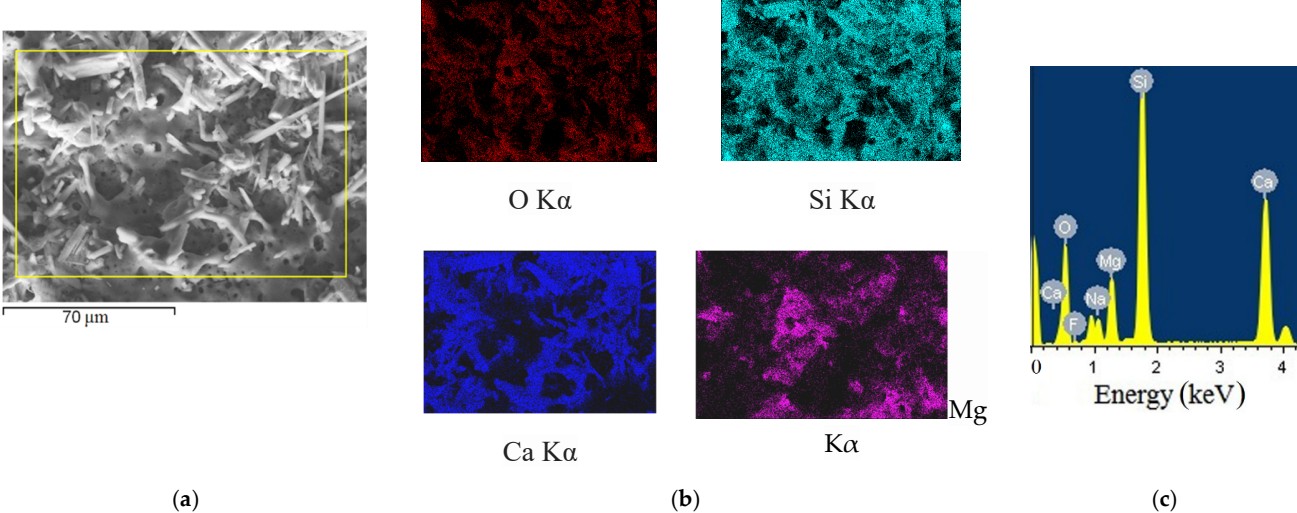

**Figure 4.** SEM image of the W-coating deposited at 450 V (**a**), element distribution maps (**b**), and the energy spectrum (**c**).

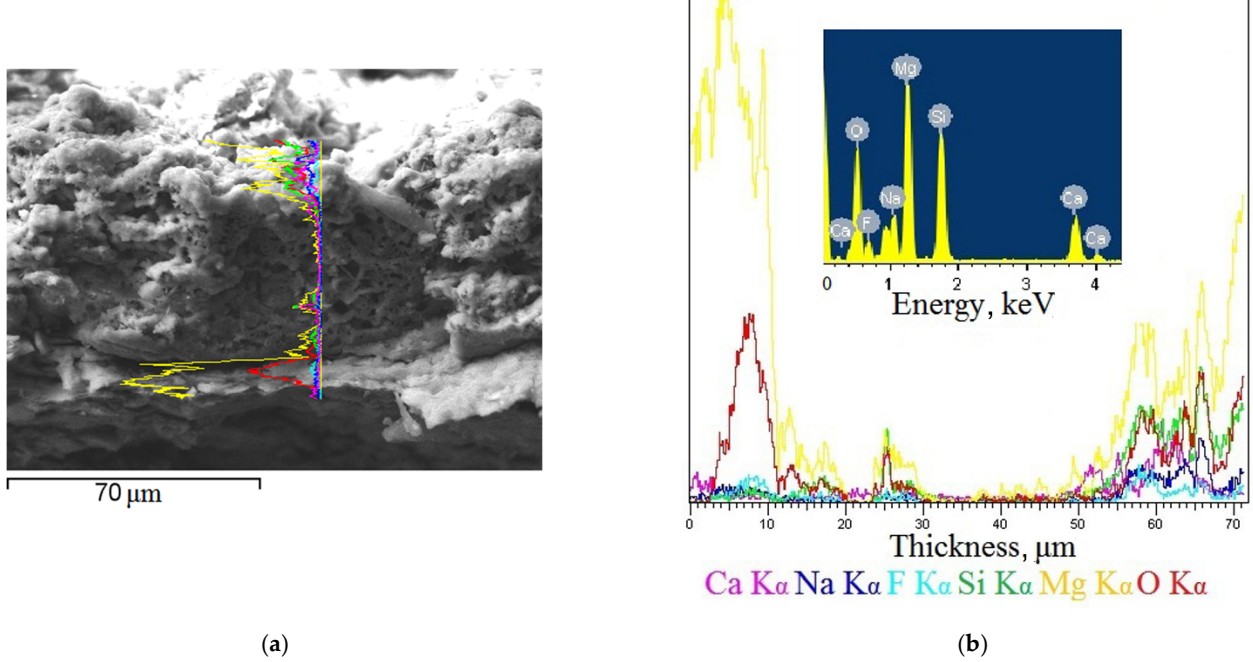

**Figure 5.** SEM image of the cross-section of the W-coating deposited at 450 V (**a**), element distribution tracks through the W-coating's thickness (**a**,**b**), and the energy spectrum (**b**).

The energy spectrum (Figure 5b) shows that Mg, Si, and O are the predominant elements in the cross-section of the coating, while the amount of Ca is lower inside the coating than on the coating's surface. Fluorine is also present, mostly in the coating's cross-section, because the $MgF_2$ was formed together with the oxide in the protective layer

between the substrate and the coating. The presence of sodium was revealed in the coatings. It is possible that it diffused from the electrolyte and embedded into the amorphous phase of the coatings.

Increasing the applied voltage led to an increase in the amount of Ca, while the amount of Mg decreased both in the surface and in the cross-section of the coatings (Table 1). This indicates that with the increase in voltage a greater amount of the wollastonite was involved in the coating synthesis process.

**Table 1.** The elemental composition of the W-coatings on Mg0.8Ca is determined by the EDX method.

| Elements | Content of Elements (at. %) | | | | | | | |
| | On the Surface | | | | In the Cross-Section | | | |
| | 350 V | 400 V | 450 V | 500 V | 350 V | 400 V | 450 V | 500 V |
|---|---|---|---|---|---|---|---|---|
| O | 63.8 | 64.1 | 63.3 | 62.1 | 60.3 | 60.0 | 61.0 | 60.2 |
| Mg | 9.5 | 7.9 | 6.3 | 4.1 | 26.4 | 24.0 | 21.3 | 19.1 |
| Si | 17.8 | 17.2 | 17.5 | 16.9 | 7.5 | 9.3 | 10.0 | 11.8 |
| Ca | 6.3 | 7.8 | 9.9 | 13.6 | 2.5 | 3.2 | 3.8 | 4.9 |
| Na | 2.1 | 2.3 | 2.2 | 2.5 | 1.8 | 1.9 | 2.1 | 2.3 |
| F | 0.5 | 0.7 | 0.8 | 0.8 | 1.5 | 1.6 | 1.8 | 1.7 |

### 3.2. Phase Composition of the W-Coatings

The XRD results demonstrated that wollastonite was the main crystalline phase (Figure 6). Reflexes of wollastonite in the XRD patterns of the coatings were almost exactly repeated from the reflexes observed in the spectrum of the initial wollastonite (Figure 6b). In addition, magnesium oxide (periclase) and magnesium silicate (forsterite) were identified in the coatings. The formation of the oxide layer took place at the initial moment of the micro-arc oxidation process, and was located at the boundary between the substrate and the coating. This fact is confirmed by the results of the elemental analysis (Figure 5). Forsterite was formed by the interaction of silicate ions with a magnesium substrate, according to the reaction (3) [59].

$$2Mg^{2+} + SiO_3^{2-} + 2OH^- \rightarrow Mg_2SiO_4 + H_2O \tag{3}$$

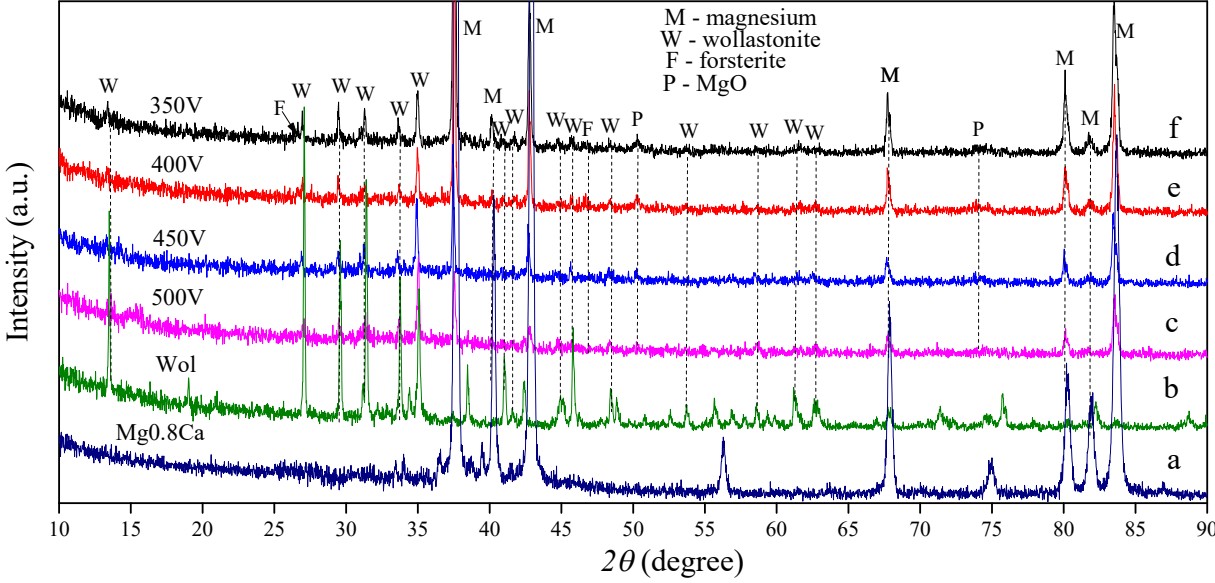

**Figure 6.** X-ray powder diffraction (XRD) patterns of the Mg0.8Ca alloy (**a**), the wollastonite (**b**), and the W-coatings deposited at the different voltages (**c**–**f**).

With an increase in the applied voltage, the reflexes of periclase practically disappeared, and the intensity of the magnesium reflexes became lower. A larger amount of the amorphous phase appeared in the coatings, as evidenced by an increase in the halo.

Studies of the coating's structure were carried out using the Rietveld method [62,63]. The volume ratio of the crystalline and amorphous phases was calculated (Figure 7). A fundamental change in the W-coating's structure from crystalline to amorphous–crystalline occurred when the voltage increased from 350 V to 400 V. As Figure 2a demonstrates, this may be due to the fact that in this case the initial current density has almost doubled.

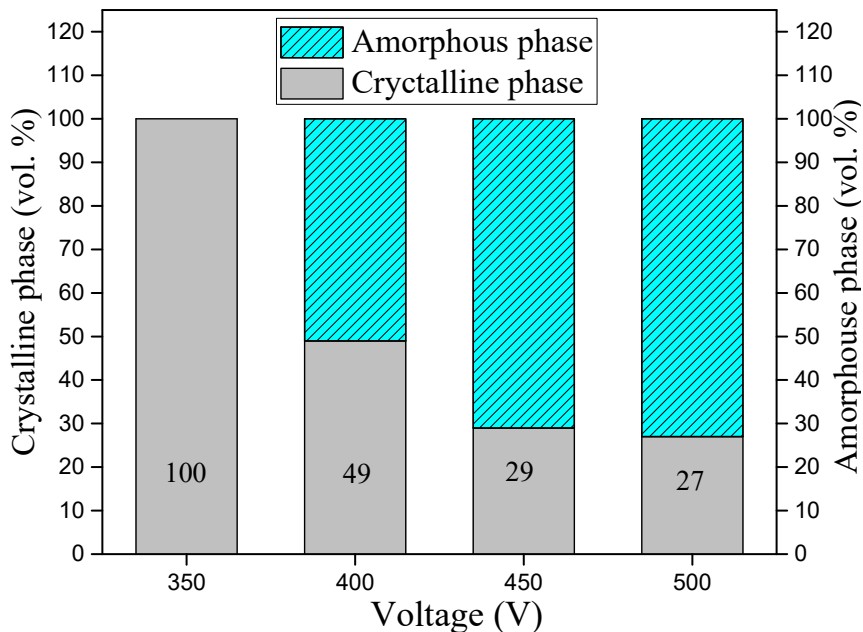

**Figure 7.** Diagram of the ratio of amorphous and crystalline phases in the W-coatings.

Accordingly, the intensity of the micro-arc discharges increased, which led to the melting and rapid solidification of the coating substance in the form of the amorphous–crystalline layer [64]. The XRD results consistent with the data were obtained via the SEM method (Figure 3).

Through the useof transmission electron microscopy (TEM), selected area diffraction (SAD) patterns, light-field, and dark-field TEM images of the fragments of W-coatings deposited at the different voltages were obtained (Figure 8).

The presence of crystalline phases of wollastonite (JCPDS No. 43-1460) and forsterite (JCPDS No. 34-0189) in the coatings was confirmed (Figure 8a,c). In addition, akermanite crystallites ($Ca_2MgSi_2O_7$) (JCPDS No. 35-0592) were found. The sizes of akermanite crystallites are observed in the reflections (201) (Figure 8b), and (630) (Figure 8d) were 200–300 μm. Akermanite crystallites are located mainly on the surface of wollastonite particles. Therefore, it is assumed that the formation of akermanite occurred at the interface when wollastonite interacted with magnesium, according to the reaction (4).

$$2CaSiO_3 + Mg^{2+} + 2OH^- \rightarrow Ca_2MgSi_2O_7 + H_2O \tag{4}$$

On the phase diagram of the three-component system CaO–MgO–SiO$_2$ [58], the crystallization fields of wollastonite and akermanite, as well as those of akermanite and forsterite, border one another, confirming the possibility of the formation and simultaneous presence of these compounds in the coatings.

Figure 9 shows the IR spectra of the wollastonite (a) and W-coatings formed at voltages of 350 and 500 V (b).

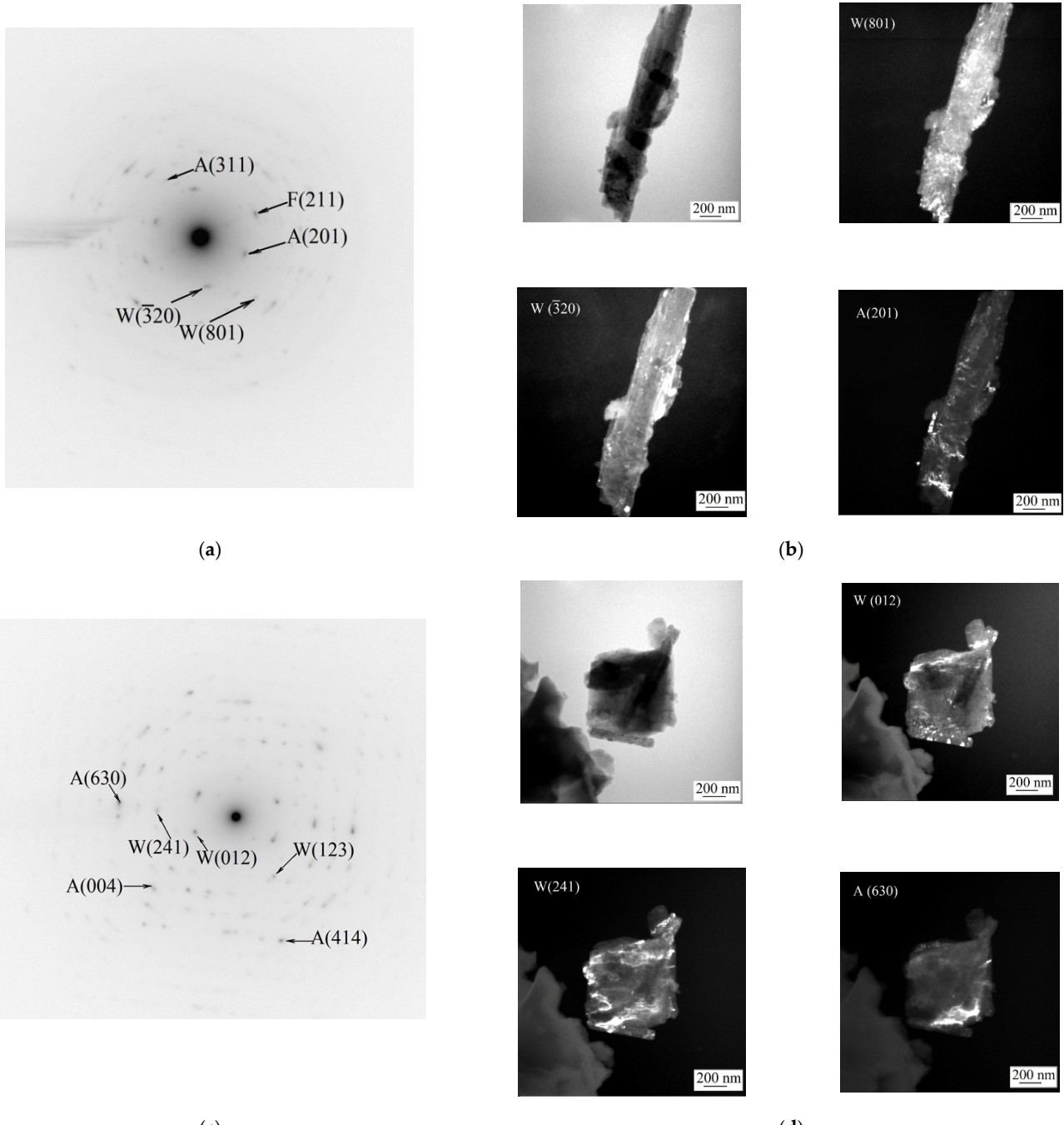

**Figure 8.** SAD pattern (**a**,**c**), light-field, and dark-field (**b**,**d**) TEM images of the fragments of W-coatings deposited at the voltages of 350 V (**a**,**b**) and 500 V (**c**,**d**).

In the FTIR spectra of natural wollastonite, the IR absorption peaks were observed in the region of 1056–1060 cm$^{-1}$, corresponding to the asymmetric stretching mode of Si–O–Si. The IR absorption peaks located at 960 and 896 cm$^{-1}$ can be connected with the non-bridging silicon–oxygen bond of Si–O. The absorption bands located at 469 cm$^{-1}$ correspond to the bending vibrational mode of Si–O–Si [32]. In addition, the bands of Si–O bonds' vibrations are superimposed on the non-bridging bonds of Ca–O in the range of 450–500 cm$^{-1}$ [65]. In the FTIR spectra of metasilicates, the region of 750–550 cm$^{-1}$ is of the greatest interest. The number of bands in this region makes it possible to judge the number of silicon–oxygen tetrahedrons in the chain recurrence period [66]. The recurrence period in the wollastonite chain is three tetrahedrons (Figure 1a), and three bands are observed in

the IR spectrum: they are 681, 642, and 564 cm$^{-1}$, respectively. Figure 9b demonstrates a decrease in intensity and broadening of the main bands in the IR spectra of the W-coatings. Moreover, if the bands corresponding to the vibrations of bonds in the wollastonite chain are still preserved for the coating formed at 350 V, these bands are no longer present in the spectrum of the coating deposited at 500 V. This indicates a distortion of the crystalline structure and its partial amorphization.

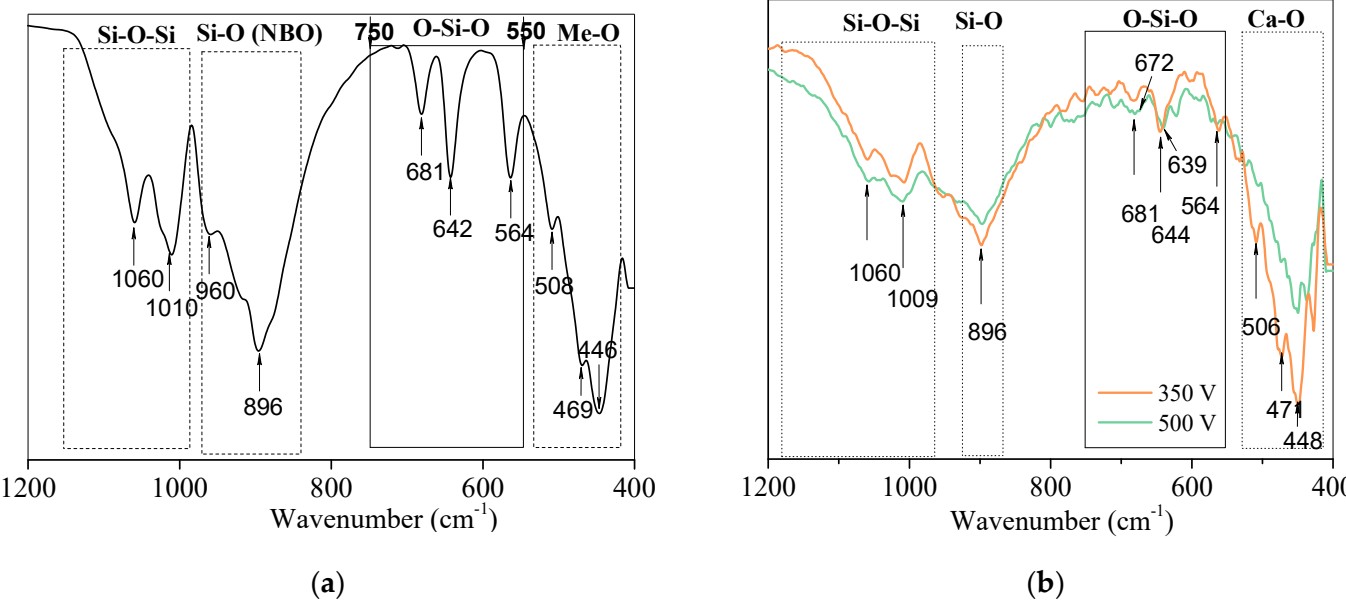

**Figure 9.** FTIR spectra of the wollastonite (**a**) and of the W-coatings deposited at the voltages 350 V and 500 V (**b**).

### 3.3. Bioresorption of the Pure Mg0.8Ca Alloy and W-Coatings

The samples of the pure Mg0.8Ca and of the coated alloy were immersed in the 0.9% NaCl solution at 37 °C for 16 days. It was revealed that the dissolution rate of the coated samples was significantly lower than that of the pure magnesium alloy (Figure 10a).

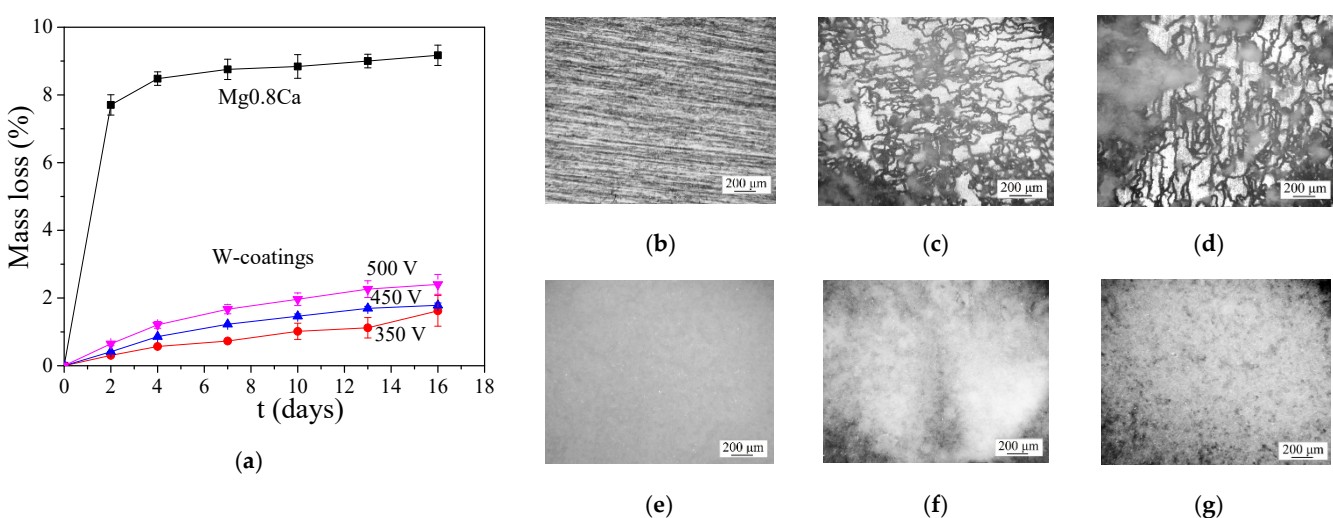

**Figure 10.** Graphs of the mass loss over time (**a**); optical images of the bare Mg0.8Ca alloy (**b**–**d**) and of the W-coating deposited at the voltage 500 V (**e**–**g**); (**b**,**e**)—initial samples, (**c**,**f**)—after 7-day dissolution, (**d**,**g**)—after 16-day dissolution in 0.9% NaCl.

Figure 10b–g shows the optical images of the Mg0.8Ca and of the coated sample before

the dissolution, after 7-day dissolution, and after 16-day dissolution in 0.9% NaCl. The pure magnesium alloy dissolved explosively. The deposition of dissolution products was observed on the surface of the magnesium alloy as a white bloom, despite the intensive bioresorption of the samples (Figure 10e–g). The coating retained its integrity for 16 days (Figure 10b–d). No large cracks or deep corrosion pits were observed in the optical images of the coating surface.

SEM images of the alloy surface and of the W-coating after 16-day dissolution are shown in Figure 11. Lamellar crystals forming into rosettes are observed at high magnification on the surface of the magnesium alloy.

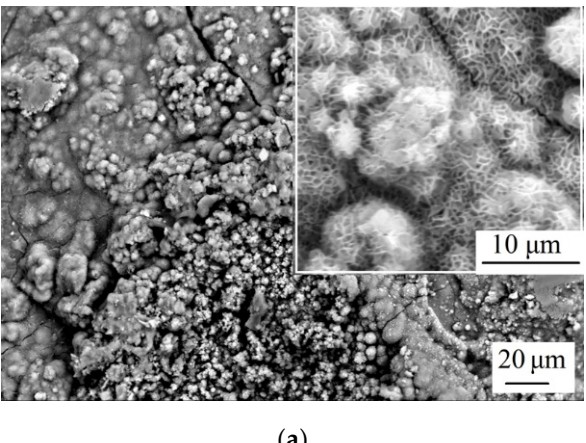

(**a**)

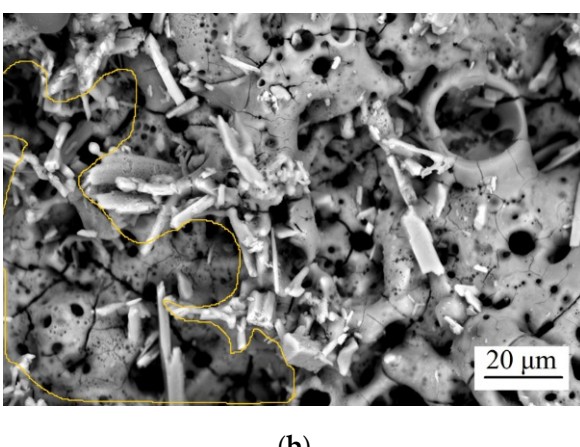

(**b**)

**Figure 11.** SEM images of the surface of the Mg0.8Ca alloy (**a**) and of the W-coating (**b**) after 16-day dissolution in 0.9% NaCl.

In the XRD patterns (Figure 12a) it can be seen that the intensity of magnesium reflexes decreased significantly after bioresorption, and some new reflexes appeared. These reflexes are related to the crystalline phases of $Mg(OH)_2$ and $MgCO_3$. TEM results confirmed the phase composition of the precipitate deposited on the magnesium alloy (Figure 12b). Light-field TEM images show that the diameter of the lamellar crystals is 100 nm and their thickness is 20 nm.

The dissolution of the coating occurred more intensively in the regions free from the wollastonite particles (Figure 11b). In the SEM image, a yellow line marks the area of the coating's dissolution. The pores and cracks have become deeper; however, the crystals of the wollastonite retained their characteristic elongated shape.

The comparative analysis of the XRD patterns of the coating before and after dissolution (Figure 12c) shows that the intensity of the wollastonite reflexes decreases while the intensity of those of the magnesium substrate increases. This means that the wollastonite crystals also participated in the dissolution process.

The energy EDX spectra of the W-coating (Figure 12d) demonstrate that after dissolution the amount of silicon and calcium decreased, while the amount of magnesium increased. In this case, as a result of the dissolution, the W-coating became thinner and the magnesium substrate shone through more intensely.

### 3.4. Electrochemical Properties

To determine the electrochemical properties of the bare magnesium alloy and the coated samples, potentiodynamic curves were obtained and electrochemical parameters were calculated. The polarization curve of the alloy Mg0.8Ca is typical for this type of material (Figure 13a) [27,67]. For the samples with W-coatings, polarization curves were in the field of lower currents compared to the curve for bare Mg alloy. In addition, significant suppression of anodic and cathodic reactions was found.

Electrochemical parameters are presented in Table 2. The corrosion current values for the coated samples were almost ten times lower than that for the Mg0.8Ca. At the same

time, the corrosion resistance values of the coatings were by an order of magnitude—or even two orders of magnitude—higher than that of the magnesium alloy. The W-coating deposited at the voltage of 500 V was the most resistant, because it had the lowest corrosion current value. In this case, the polarization resistance value and impedance modulus were the highest.

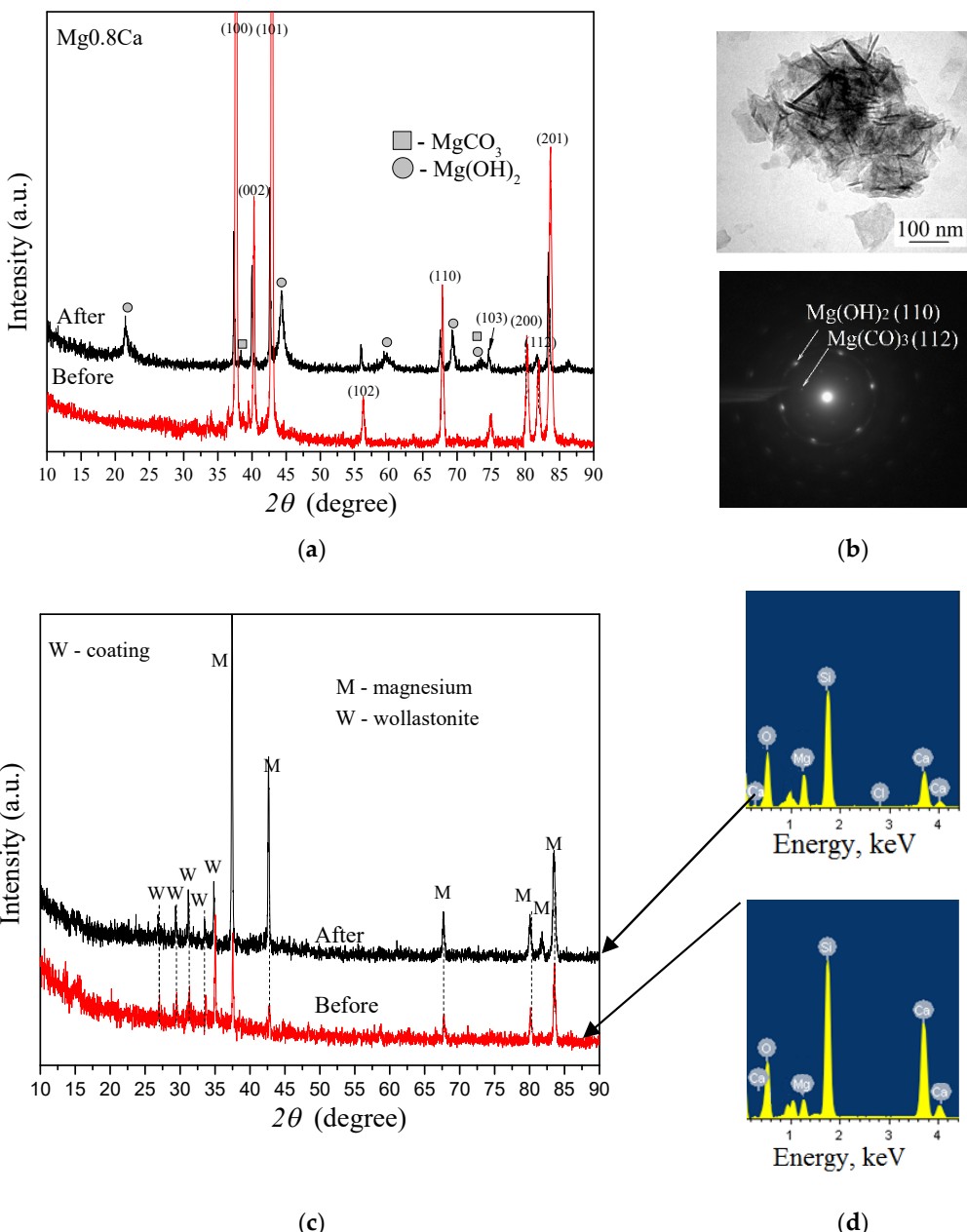

**Figure 12.** XRD patterns (**a,c**), SAD pattern and light-field TEM image (**b**), EDX energy spectra (**d**) of the Mg0.8Ca alloy (**a,b**) and of the W-coatings (**c,d**) before and after 16-day dissolution in 0.9% NaCl.

It should be mentioned that after the formation of the coating at 350 V the corrosion potential value shifts to lesser values in comparison with the bare alloy, along with an increase in polarization resistance. This can be explained as follows: The coating had complicated phase composition and developed surface morphology. Some of the soluble compounds in the coating composition can interact with the NaCl solution, changing the potential-determining reaction. At the same time, this coating had a thickness of 40 μm, and demonstrated high corrosion resistance, shielding the metal surface from contact

with the aggressive medium. The coatings formed at 400 and 500 V are characterized by slightly nobler values of corrosion potential, due to the transition from crystalline to crystalline–amorphous structure, and more pronounced barrier properties explained by their greater thickness.

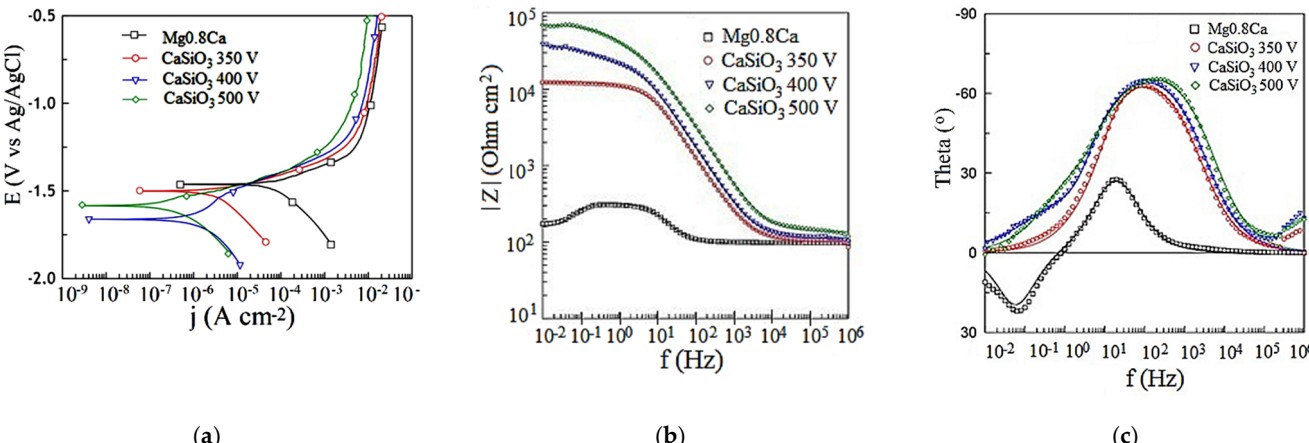

| (a) | (b) | (c) |

**Figure 13.** Potentiodynamic polarization (PDP) curves (**a**), Bode plots (the impedance modulus |Z| (**b**) and the phase angle *theta* on the frequency (**c**) dependencies) obtained in 0.9% NaCl for the bare Mg0.8Ca alloy and the W-coatings.

**Table 2.** Electrochemical parameters of the bare Mg0.8Ca alloy and the W-coatings.

| Sample | Applied Voltage, V | $E_c$, V (vs. Ag/AgCl) | $J_c$, A cm$^{-2}$ | $R_p$, Ω cm$^2$ | $Z|_{f\to 0\,Hz}$, Ω cm$^2$ |
|---|---|---|---|---|---|
| Mg0.8Ca | - | −1.33 | $7.3 \times 10^{-6}$ | $4.2 \times 10^3$ | $2.1 \times 10^2$ |
| 1 | 350 | −1.45 | $4.7 \times 10^{-7}$ | $5.1 \times 10^4$ | $1.3 \times 10^4$ |
| 2 | 400 | −1.42 | $5.2 \times 10^{-7}$ | $4.3 \times 10^4$ | $3.9 \times 10^4$ |
| 3 | 500 | −1.41 | $1.9 \times 10^{-7}$ | $1.3 \times 10^5$ | $6.8 \times 10^4$ |

The Bode plots (Figure 13b,c) show the results of impedance measurements in the form of dependences of the impedance modulus |Z| and phase angle *theta* versus the frequency.

The presence of a thin film of oxide or hydroxide on the surface of the magnesium alloy sample has an effect on the dependence of the phase angle on the frequency in the middle and low ranges. The value of the impedance modulus $|Z|_{f\to 0\,Hz}$ of Mg0.8Ca in the low-frequency region was equal to $2 \times 10^2$ Ω cm$^2$ (Table 2). Thus, the bare magnesium alloy is highly reactive, and the protective coatings are necessary in order to increase its corrosion resistance.

For modelling the impedance spectra, the equivalent electrical circuits (EEC) for the Mg0.8Ca alloy and W-coatings were created (Figure 14). To demonstrate the samples' electrochemical behavior, the CPE were used in the EEC:

$$Z_{CPE} = 1/[Q(j\omega)^n] \tag{5}$$

where $\omega$ is an angular frequency ($\omega = 2\pi f$), $j$ is an imaginary unit, and n and $Q$ are the exponential coefficient and the frequency independent constant, respectively. In this case, $n$ shows the deviation from ideal capacitive behavior. When $n = 0$ and 1 the physical meaning of $Q$ became classic elements of conductivity (1/R) and capacitance (C), respectively.

The presence of capacitive in the middle and inductive in the low frequency range loops was revealed for the bare Mg0.8Ca and for the coated samples. The $R_1$–$CPE_1$ capacitive loop for Mg0.8Ca was connected with the charge transfer resistance and the capacitance of the electrical double layer (Figure 14a). For the samples with W-coatings, the R1–$CPE_1$ capacitive loop corresponded to the electrolyte resistance in the pores and the geometrical capacitance of the oxide layer, whereas $R_2$–$CPE_2$ exhibited parameters of

the barrier layer of the W-coating (Figure 14b). The parameters of the EECs elements are presented in Table 3.

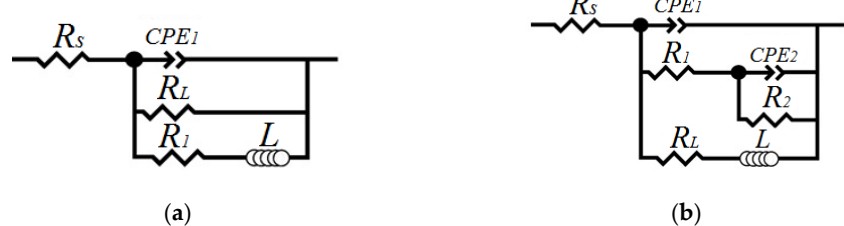

**Figure 14.** Equivalent electrical circuits used for modelling the impedance spectra for: (**a**)—bare Mg0.8Ca alloy, and (**b**)—W-coatings.

**Table 3.** Parameters of the EECs elements.

| Sample | $CPE_1$ | | $R_1$ ($\Omega$ cm$^2$) | $CPE_2$ | | $R_2$ ($\Omega$ cm$^2$) | $R_L$ ($\Omega$ cm$^2$) | $L$ (H cm$^2$) |
|---|---|---|---|---|---|---|---|---|
| | $Q_1$ (S cm$^{-2}$ s$^n$) | $n$ | | $Q_2$ (S cm$^{-2}$ s$^n$) | $n$ | | | |
| Bare Mg0.8Ca | $1.1 \times 10^{-4}$ | 0.91 | 217.1 | - | - | - | 78.0 | 555.2 |
| 350 V | $5.1 \times 10^{-6}$ | 0.78 | $1.7 \times 10^4$ | $4.1 \times 10^{-5}$ | 0.48 | $1.3 \times 10^3$ | $3.8 \times 10^4$ | 524.7 |
| 400 V | $3.4 \times 10^{-6}$ | 0.79 | $2.2 \times 10^4$ | $2.1 \times 10^{-5}$ | 0.57 | $5.6 \times 10^4$ | $7.9 \times 10^4$ | 4372.0 |
| 500 V | $1.8 \times 10^{-6}$ | 0.80 | $3.6 \times 10^4$ | $5.3 \times 10^{-6}$ | 0.68 | $8.4 \times 10^4$ | $1.7 \times 10^5$ | 5423.0 |

The $R_L$–L chain was connected with the relaxation processes that took place during the dissolution of magnesium and deposition of corrosion products on the phase boundary of the "alloy/electrolyte" interface for untreated samples, and of the "alloy/coating/electrolyte" interface for the coated samples [68].

Thus, the W-coatings demonstrated the ability to reduce the corrosion rate of the magnesium alloy and showed excellent corrosion resistance, especially those formed at 500 V.

### 3.5. Biological Research

The study of the effect of the sample extracts on 3T3 fibroblast cell viability showed that when cells contacted the extracts of the W-coating and of the bare magnesium alloy, the number of surviving cells was equal to 85% and 28% respectively (Figure 15a).

When the coating extracts were diluted 10 and 100 times, the number of surviving cells increased to 97%, and approached the control. In this case, dilution of extracts from the pure magnesium samples resulted in a threefold increase in surviving cells, but their number remained low in comparison with coated samples (Figure 15a). The sample extracts were diluted to simulate the flow conditions of biological fluids in the body.

The performed studies showed that the coated samples did not greatly affect the viability of the cell line, even without additional dilution of the extracts, and were non-toxic according to ISO 10993-5: 2009 (the decrease in viability does not exceed 20% relative to the negative control). Thus, it was found that the W-coatings significantly reduce toxicity of the magnesium alloy.

An important condition for maintaining cell viability is the maintenance of the medium's pH in the range of neutral values. When the test samples were immersed in a cell culture medium, a change in pH was observed (Figure 15b). Hydrogen evolution during the dissolution of the magnesium alloy led to an increase in the pH of the cell medium to 9.8 within 25 h of exposure. The coating helped to reduce the pH level, which prevented the medium from alkalizing in the implantation site (Figure 15b). The obtained data correlated well with the toxicity data.

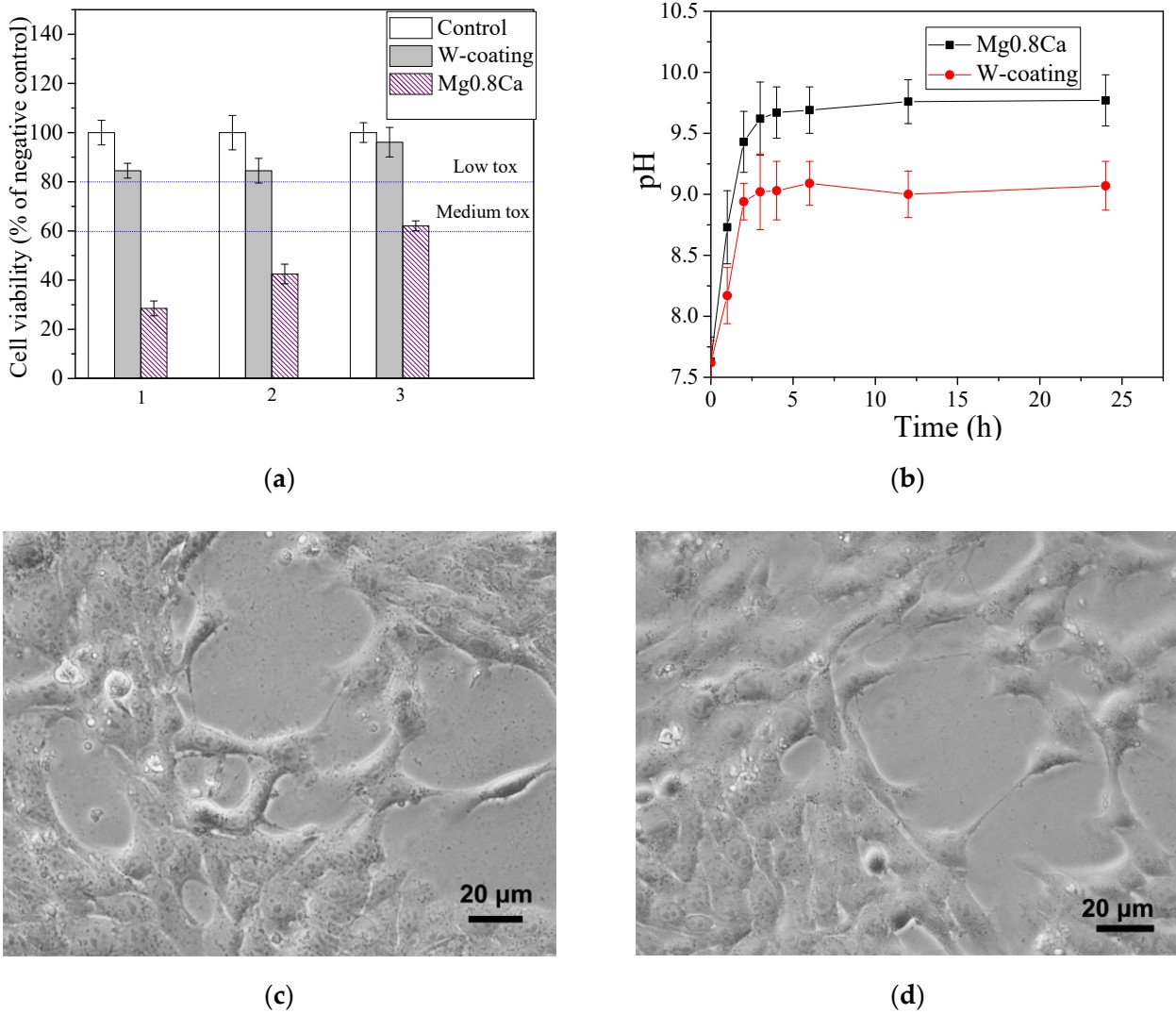

**Figure 15.** (**a**) — Results of the in vitro cytotoxic test of the Mg.08Ca and W-coating extracts measured by an MTT assay (1); when the extracts were diluted 10 (2) and 100 (3) times. (**b**)—Kinetics of pH changes in the cell medium during interaction with the samples. (**c**,**d**)—3T3 cells after incubation with the control and with the coating, respectively.

Optical microscopy enables us to assess the changes in the cell morphology after their contact with the W-coating extracts. As shown in Figure 15c,d, the cells cultured in plates in the presence of the sample extract did not differ morphologically from the cells in the control group cultured only in nutrient medium. Thus, the extracts of the samples did not cause changes to the morphology of the 3T3 cell line.

## 4. Discussion

The correlation of the analysis of the elemental- and structural-phase composition of the coatings, investigated using the EDX, XRD, TEM, and IR methods, suggests the following: Owing to the peculiarities of the micro-arc oxidation process in the unipolar potentiostatic mode in an electrolyte containing both solutes and a dispersed phase, the coatings were formed with a specific structure. At the initial moment of the MAO process (Figure 2a), the current density reached its highest value. The most powerful micro-arc discharges occurred at the interface between the metal substrate (oxide film) and the electrolyte. This led to intense melting of the electrolyte substance in the micro-arc discharge channel, and the formation of a porous amorphous–crystalline layer. As Khan et al. [64] reported, because of the action of short-lived micro-discharges, non-stationary processes take place, such as a very fast heating/cooling, thermochemical interaction

between the substrate and the electrolyte, instant melting, and the subsequent solidification of the coating.

Within 50 s of the coatings' formation, the intensity of micro-arc discharges decreased (Figure 2a). At the final stage, when the current density reached its minimal value and the micro-arc discharges were extinguished, wollastonite particles were deposited on the coating's surface (Figure 16). Previous studies have also shown that when the coatings were formed via the MAO method, using a potentiostatic mode in an electrolyte containing dispersed particles, these particles were transferred from the electrolyte to the coating's surface [27,67]. Thus, it is assumed that the main crystalline phase of wollastonite was presented on the coating surface. In addition, small quantities of akermanite nanocrystallites were formed on the wollastonite particles, but they were detected only by TEM.

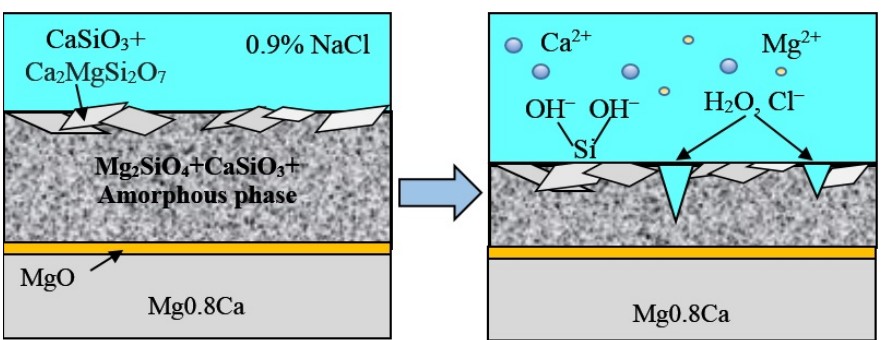

**Figure 16.** Schematic image of the coating dissolution process in 0.9% NaCl solution.

Magnesium silicate, forsterite, and magnesium oxide were mainly formed inside the coating, since Mg, Si, and O prevailed in the elemental composition of the coating's cross-section. The authors of [69,70] wrote about the formation of forsterite and periclase in the interaction of the electrolyte containing sodium silicate with a magnesium substrate. Furthermore, as was revealed using the SEM EDX method, in addition to the oxide, $MgF_2$ also formed in the boundary protective layer between the coating and the substrate. Since a halo was observed in the XRD pattern of the coatings, one can conclude that the coatings contained an amorphous phase. Moreover, with an increase in the voltage up to 500 V, the amount of the amorphous phase in the coatings' phase composition increased dramatically. The dissolution of the coating in the solution of 0.9% NaCl occurred most intensively in the regions between the crystals, where the amorphous phase was predominant (Figure 16).

On the other hand, with an increase in the process voltage, the amount of calcium in the coatings rises, which indicates an increase in the involvement of wollastonite in the MAO process. The corrosion resistance of the coatings increased in this case, because it was found that coatings applied at a process voltage of 500 V had the highest corrosion resistance. Sainz et al. [34] described the dissolution process of wollastonite in SBF. The reaction (6) took place in this case:

$$CaSiO_3 \text{ (s)} + H_2O \rightarrow Ca^{2+} \text{ (aq.)} + HSiO_3^- \text{ (aq.)} + OH^- \text{ (aq.)} \tag{6}$$

When wollastonite dissolved, calcium ions were released from the surface. In this case, many silanol groups (Si–OH) were formed. These groups were the centers of the formation and growth of hydroxyapatite [28,29]. The released calcium ions, in turn, intensified apatite nucleation. Similar processes occurred during the dissolution of Ca- and Mg-silicates, such as akermanite [39] and forsterite [69].

Zakaria et al. [30] showed that the wollastonite coating had good biological activity, both in the in vitro tests with a model of biological fluid, and in the in vivo tests on real implants for dogs. The mechanism of HA growth on the surface of a wollastonite coating was presented. Zhai et al. [37] reported that bioceramics in the system Ca–Mg–Si have a stimulating effect on osteogenesis and angiogenesis. Dissolved Ca ions can induce

osteoblast proliferation, while Mg stimulates the mineralization of calcified tissues and affects mineral metabolism [71].

In the presented work, the wollastonite contained in the coatings contributed to the improvement of the biocompatibility of the magnesium samples, since there was a significant difference in the number of viable cells after interacting with the bare magnesium alloy and with the coated samples.

## 5. Conclusions

Porous coatings containing crystalline phases such as wollastonite, akermanite, forsterite, magnesium oxide, and amorphous phase were synthesized on the surface of the Mg0.8Ca alloy by means of the MAO method.

Elongated crystals of wollastonite were uniformly distributed over the surface of the coatings. It is assumed that akermanite was formed as a result of the interaction of wollastonite with the magnesium substrate during the micro-arc discharges, while forsterite was formed when the magnesium substrate interacted with sodium silicate contained in the electrolyte.

The thickness and roughness of the coatings increased from 40 to 150 μm, and from 6.5 to 10.5 μm, respectively, with an increase in the process voltage in the range of 350–500 V. The rate of dissolution of the coated samples was significantly lower than that of the pure magnesium alloy. The corrosion current of the coated Mg0.8Ca decreased ten times, and its corrosion resistance increased almost a hundred times.

Thus, it was shown that due to the special structure of the micro-arc coatings, their complex amorphous–crystalline structure, and their phase composition, the W-coatings exhibited significantly low bioresorption rates and remarkable corrosion resistance. In addition, the W-coatings were not cytotoxic to the 3T3 fibroblast cell line, and improved the biocompatibility of the magnesium alloy.

The developed biodegradable biocomposite based on the Mg0.8Ca alloy and the W-coating is a promising material for use in implants in reconstructive medicine.

**Author Contributions:** Conceptualization, M.B.S. and Y.P.S.; methodology, A.V.U.; software, A.V.U.; validation, M.B.S., A.V.U. and O.V.B.; formal analysis, V.V.C., I.A.C.; investigation, T.V.T., V.V.C. and I.A.C.; resources, J.S.; data curation, T.V.T., M.A.K. and V.S.E.; writing—original draft preparation, M.B.S.; writing—review and editing, M.B.S., O.V.B., and V.S.E.; visualization, V.S.E. and J.S.; supervision, O.V.B.; project administration, M.I.L.; funding acquisition, M.I.L., J.S. and Y.P.S. All authors have read and agreed to the published version of the manuscript.

**Funding:** The work was performed according to the Government research assignment for ISPMS SB RAS, project FWRW-2021-0007.

**Acknowledgments:** The authors express sincere appreciation for the valuable contributions of A.I. Tolmachev from Institute of Strength Physics and Materials Science SB RAS (Tomsk, Russia) for his assistance in the preparation of the experimental materials.

**Conflicts of Interest:** The authors declare no conflict of interest.

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
