# Peer review of "Surface Modification of Mg0.8Ca Alloy via Wollastonite Micro-Arc Coatings: Significant Improvement in Corrosion Resistance"

_metals, doi:10.3390/met11050754_

Round 1

Reviewer 1 Report

1、The voltage increases and the amount of wollastonite used increases. Why does the amount of calcium increase and the amount of magnesium decrease?

2、The EDX spectrum shows that the magnesium content increases. The content increase is due to the dissolution of the coating or the corrosion of the magnesium matrix?

3、Although the coating has lowered the pH value, the pH value is still around 9 in 25 hours, Does it has any effects on cell behaviors?

4、In cell viability experiments, does the cell culture time affect the number of viable cells?

5、No results of cell adhesion and spread can be found on the coating and pure magnesium samples.

6、The author should provide the result of cell growth in MEM medium。

7、Does the author provide OD values at 570nm?

Reviewer 2 Report

In the submitted manuscript, the author prepared a kind of protective silicate biocoatings on Mg0.8Ca alloy by means of the micro-arc oxidation method and studied their morphology, structure, phase composition, corrosion and biological properties .The coated samples exhibited remarkable anti-corrosion properties and applied coatings significantly reduced the toxicity of the alloy in comparison with bare magnesium alloy.However, there are still some contents that need to be checked and modified in the manuscript.

  1. The introductionprovides insufficient background, such as examples of applications of Micro-arc oxidation(MAO) .
  2. The results show that the coating has the best performance at 500V.Does the author consider how the performance will change as the voltage continues to increase?
  3. Does the author consider the effect of the time of MAO on the coating properties ?
  4. The date is not clearly expressed in Figure 15(b).

Reviewer 3 Report

The paper is suitable for the journal Metals  and topical for nowadays when the biodegradable alloys are more investigated for their  great ability to improve the properties  after surface modification such as micro-arc wollastonite coating.

In order to be published my recommendation is major revision introducing details about novel character in comparison with other papers such as two conference papers in 2019 entitled „ Corrosion behavior of the Mg0.8Ca alloy modified by wollastonite coating. IP Conference Proceedings, 2019, 2167, 020314 ” and „Formation and properties of micro-arc silicate biocoatings on bioresorbable alloy Mg0.8Ca” IOP Conf. Series: Journal of Physics: Conf. Series 1281 (2019) 012070 IOP Publishing doi:10.1088/1742-6596/1281/1/012070 which are devoted  to wollastonite micro-arc coating formation and characterization. 

The biological part has to be extended   the MTT test with some comments  being  not enough to conclude about  the effect of wollastonite coating in significantly reduce toxicity of magnesium alloy.  I do suggest  NO, ALP,ROS or other experiments to enrich this part and manuscript novelty.

Round 2

Reviewer 1 Report

The authors solve all my concerns, so I would like to recommend for publication. 

Author Response

The Authors are very grateful to the  Reviewer for a positive assessment of the manuscript.

Reviewer 3 Report

In my opinion I do believe that the changes introduced after the second review does not represent significantly improvement  and is a need to really introduced new experiments to enhance novel character  and scientiific level of the paper. The  real important part of the work was aldready presented previously and small modifications  were not able to introduce novelty.

It is still a need for major revision
